# RNAseq-Based Working Model for Transcriptional Regulation of Crosstalk between Simultaneous Abiotic UV-B and Biotic Stresses in Plants

**DOI:** 10.3390/genes14020240

**Published:** 2023-01-17

**Authors:** Zheng Zhou, Alex Windhorst, Dirk Schenke, Daguang Cai

**Affiliations:** 1Department of Molecular Phytopathology and Biotechnology, Institute of Phytopathology, Christian-Albrechts-Universität zu Kiel, 24118 Kiel, Germany; 2Plant Breeding Methodology, Georg-August-University Göttingen, 37073 Göttingen, Germany

**Keywords:** flg22, PTI, MBW complex, transcriptomics, transcription factor

## Abstract

Plants adjust their secondary metabolism by altering the expression of corresponding genes to cope with both abiotic and biotic stresses. In the case of UV-B radiation, plants produce protective flavonoids; however, this reaction is impeded during pattern-triggered immunity (PTI) induced by pathogens. Pathogen attack can be mimicked by the application of microbial associated molecular patterns (e.g., flg22) to study crosstalk between PTI and UV-B-induced signaling pathways. Switching from Arabidopsis cell cultures to in planta studies, we analyzed whole transcriptome changes to gain a deeper insight into crosstalk regulation. We performed a comparative transcriptomic analysis by RNAseq with four distinct mRNA libraries and identified 10778, 13620, and 11294 genes, which were differentially expressed after flg22, UV-B, and stress co-treatment, respectively. Focusing on genes being either co-regulated with the UV-B inducible marker gene chalcone synthase *CHS* or the flg22 inducible marker gene *FRK1* identified a large set of transcription factors from diverse families, such as MYB, WRKY, or NAC. These data provide a global view of transcriptomic reprogramming during this crosstalk and constitute a valuable dataset for further deciphering the underlying regulatory mechanism(s), which appear to be much more complex than previously anticipated. The possible involvement of MBW complexes in this context is discussed.

## 1. Introduction

Plants respond to environmental cues with a plethora of induced defense reactions involving changes in secondary metabolite composition. In light of increased stress situations due to climate change, it is anticipated that combinations of abiotic and biotic stresses will pose a challenge to agricultural production. Therefore, it is important to understand the underlying molecular mechanisms of how plants perceive stress and integrate different stress signals in order to launch an appropriate defense to ensure survival. Revealing these mechanisms will allow the identification of suitable breeding targets to create stress-resilient crops. UV-B is a well-known inducer of flavonoid production. It is known for many years that this reaction is reliably suppressed during various plant-pathogen interactions. Infection can be mimicked by the elicitation with highly conserved elicitors, which are now termed microbial associated molecular patterns (MAMPs) [1]. This very stable crosstalk can be observed in many plant species and is therefore ideal to study mechanisms on how plants integrate different stress signals [2].

Plants exposed to both UV-B and pathogens/elicitors react with the suppression of UV-B-induced genes being involved in flavonol production, regardless if MAMPs are applied concomitantly or after the onset of UV-B radiation [3,4,5,6]. The binding of MAMPs to the receptor elicits the natural pathogen defense response of plants to pathogens termed pattern-triggered immunity (PTI). One MAMP which can be artificially synthesized is the proteinaceous flg22, a 22 amino acid peptide derived from the bacterial flagellin, which binds to the pattern recognition receptor FLS2 [7]. The key enzyme of the flavonol-branch of the phenylpropanoid pathway (PP) is the chalcone synthase (CHS) constituting the entry point for flavonol and subsequent anthocyanin production [8,9] being strongly UV-B up-regulated and suppressed during PTI [5,6,10], and a perfect marker gene to investigate this crosstalk. The advantage lies in the re-distribution of resources and metabolic flow into other branches of the PP leading to the production of lignin and the coumarin phytoalexin scopoletin, components for a strong pathogen defense reaction [5]. In addition, PTI induced by flg22 goes along with strong up-regulation of *FRK1*, senescence, and flg22-induced receptor-like kinase, which can serve as a marker gene for successful plant treatment with this MAMP [11].

Plants require special protection against UV-B radiation because of their sessile nature and dependency on light. Strong UV-B radiation can damage membranes and organelles within the cell as well as DNA within the nucleus. The effect of UV-B radiation varies with the intensity and duration of irradiation and can directly or indirectly affect basic plant metabolic processes, such as respiration, photosynthesis, growth, and reproduction. At low doses as used in this study, UV-B light typically activates UV-B photoreceptor UV RESISTANCE LOCUS 8 (UVR8) [12], which will lead to a diversity of changes, such as the alternation of flowering time, reduction of hypocotyl growth, and biosynthesis of protective flavonoid metabolites [13]. In its inactive state, UVR8 forms a homodimer, which monomerizes upon UV-B perception and then translocates from the cytosol into the nucleus to interact with the E3 ubiquitin ligase COP1 [14], that is however regulated by white light [15]. In the nucleus UVR8-COP1 complex formation involves SPA1 (SUPPRESSOR OF PHY A-105 1) and this can stabilize the bZIP transcription factor (TF) HY5 by preventing its degradation [14]. HY5 is a master-regulator of the photomorphogenic UV-B responses, including the production of flavonoids [16,17]. A significant suppression of several structural genes in flavonoid biosynthesis was observed at 6 h, while at 3h the suppressing effect of flg22 was only visible by tendency, and later at 24h the effect was declining [5]. The regulation of gene expression within this crosstalk involves chromatin remodeling by preventing acetylation of histone 3 lysine residue #9 (H3K9) at the flavonol pathway gene (FPG) loci (such as *CHS*) in presence of flg22 [18]. H3K9 acetylation is a hallmark of gene activation and was affected at both the promoter and coding region of the MYB TF MYB12, a positive regulator of the CHS. Other MYB TFs were found to be differentially regulated in the context of this crosstalk with functional redundancy. For example, MYB11, 12, and 111 are positive regulators of the flavonoid pathway [19] and all are co-regulated with the marker gene *CHS* being UV-B inducible and suppressed by flg22. MYB4 and its close relatives MYB7 and MYB32 are negative regulators [20]. The *MYB4* gene was induced earlier by flg22 than by UV-B, suggesting its functional involvement in the PTI-mediated suppression of the UV-B-induced expression of FPGs [5]. Many MYBs proved to be targeted by microRNAs (miRNAs), a typical feature of TF regulation, whose expression requires frequent control by feedback circuits [21]. Moreover, MBW complexes contain MYB and basic-helix-loop-helix (bHLH) TFs as well as WD40 proteins, and proved to be implicated in the regulation of anthocyanin production [22]. All FPGs contain MYB-responsive *cis*-elements (MREs) and adjacent G-box-like sequences in their promoters [5]. In Arabidopsis and rice, the G-box binds both bZIP and bHLH TFs [23,24]. It has been assumed by Liu et al. [9] that at least some “early genes” (e.g., the *CHS*) required for flavonoid biosynthesis might be regulated by MBW complexes. 

The crosstalk as well as the postulated model merely relied on studies of in vitro Arabidopsis cell culture systems [5]. The successful establishment of a seedling-based model system [6] allowed us to investigate this crosstalk for the first time also in planta. Thus, we deployed an RNAseq experiment to gain insights into transcriptomic response in Arabidopsis and report here the identification and analysis of several TF genes, which are differentially regulated by flg22 and UV-B treatment. The large number of candidate TFs suggests different regulatory layers are implicated in this crosstalk. The results support on the one hand, our previously postulated working model, and on the other hand, demonstrate that the underlying mechanism(s) are much more complex than previously anticipated. For example, bHLH/bZIP and WD40 proteins were identified as potential interactors of the MYB TFs involved in the crosstalk. In addition, this work constitutes a valuable dataset to elucidate the molecular mechanisms underlying plant adaption to different environmental cues of biotic and abiotic origin.

## 2. Materials and Methods

### 2.1. Plant Treatment

Col-0 (WT) seedlings of *Arabidopsis thaliana* were grown on Jiffies (Jiffy-7 Peat Pellets, Jiffy Products International AS), as nine plants per Jiffy. After sowing, the seeds were placed in darkness for three days at 4 °C before being transferred to short-day conditions (8 h light, 22 °C) as previously described [6]. Five-week-old seedlings were evenly sprayed with 1 mM flg22 solution or water as a mock control. To let the effects of flg22 treatment develop, the sprayed plants were incubated for one hour in darkness and were then exposed to UV-B or VIS-light as a control for four hours. A 3 mm thick glass plate was used to constrict the UV-B radiation emitted by two broadband UV-B lamps (Philips TL 20W/12 RS) with an emission spectrum from 290 to 315 nm, as described by Rizzini et al. [12] and two PROTEC.CLASS lamps (PLSL 18W/21) for concomitant white light supply (12.69 m^2^ s^−1^) to a natural level which is sufficient to induce photomorphogenic UV-B response (0.53 µmol m^2^ s^−1^), while the control plants exposed only to VIS light were shielded with two additional layers of polyester plastic foil (Folanorm SF/AS 0.13 mm, Folex GmbH). In total three independent biological replicates were investigated consisting each of four distinct treatments: Water/VIS-light control (C), flg22 treatment/VIS-light (F), water/UV-B treatment (U), and the flg22/UV-B co-treatment (F/U).

### 2.2. Harvest and Total RNA Isolation

Immediately after the treatment, plant material was harvested and flash-frozen in liquid nitrogen for long-term storage at −80 °C. For RNA isolation the samples were homogenized with pestles in 1.5 mL tubes, placed in liquid nitrogen and total RNA was isolated by the TRIzol method. For this 1 mL of TRIzol (Invitrogen, Thermo Fisher Scientific, Waltham, MA, USA) reagent was added to each sample, it was vortexed, and incubated for 5 min at room temperature. After adding 200 µL of chloroform, vigorously vortexing, and incubation for 5 min at room temperature, the samples were centrifuged with an Eppendorf centrifuge (5417R) at 12,000 rpm for 15 min at 4 °C. The supernatant was transferred into an RNase-free 1.5 mL tube. The same volume of isopropanol (stored at −20 °C) was added and incubated for 30 min at 4 °C. Thereafter, samples were centrifuged at 12,000 rpm for 15 min at 4°C and the supernatant was discarded. The pellet was washed twice, first with 1 mL of 80 % DEPC-EtOH and then with 500 µL of 100 % EtOH. Each washing was followed by a centrifugation step at 12,000 rpm for 5 min at 4 °C and the supernatant was discarded, respectively. The pellet was dried and dissolved in 30 µL DEPC-water before being stored at −20 °C. The RNA quality and concentration were checked by agarose gel electrophoresis and with a NanoDrop (NanoVue, Biochrom) measurement, respectively. Electrophoresis was carried out using 1% agarose gels running at 100 V for 30 min.

### 2.3. cDNA Synthesis and RT-qPCR Analysis

cDNA was reversely transcribed in 20 µL volume, using the RevertAid First Strand cDNA Synthesis Kit (Thermo Fisher Scientific) according to the suppliers’ instructions with 1 µg of total RNA treated with RNase-free DNaseI (Thermo Fisher Scientific). The cDNA quality was checked by semiquantitative PCR with gene-specific primers for the reference gene *ACTIN* 2 (*ACT2*). For RT-qPCR 1 µL of cDNA was added to a 9 µL master mix consisting of a primer and SYBR green (qPCRBIO SyGreen Mix; NIPPON Genetics Europe), and qPCR was performed to analyse target gene expression relative to *ACT2* as a reference with primers described in Appendix A. The fold change in gene expression analysis was calculated with the 2^-∆∆Ct^ method [25] and multiple comparisons of statistical significance were performed by two-way ANOVA using the Minitab software [26].

### 2.4. RNAseq Analysis

In order to identify genes putatively involved in crosstalk between abiotic UV-B and biotic stress mimicked by flg22, Illumina sequencing was carried out twice with pooled RNA from three independent biological replicates comprising each the four treatments C, F, U, and F/U was carried out by Novogene Co. Ltd. (Hongkong, China). Raw reads were processed by Novogene and FPKM (Fragments Per Kilobase Million) values were used to calculate the expression level. Missing FPKM values were substituted for each gene by a default value of 0.001 when at least one FPKM value was detected in one of the four treatments (Appendix A). A weak effect of UV-B on gene expression was expected in this study since it is based on whole seedlings, not specific tissue. While leaves were directly exposed to UV-B-containing light, the roots did not receive any UV-B radiation. Furthermore, genes such as transcription factors are temporally and spatially expressed at low levels [27,28]. For these reasons, differentially expressed genes (DEGs) were calculated by applying the conventional log2 threshold of ≥1 or ≤−1 for flg22 single and co-treatments, while the log2-threshold of UV-B regulated genes was lowered to ± 0.2 in order to increase the number of these DEGs for analysis. Genes of the phenylpropanoid pathway were subjected to hierarchical clustering by using the ggplot2 package (version 3.2.1.) [29] in R [30] according to the log2 values presented in Appendix A.

### 2.5. Meta-Analysis

To further evaluate the quality of the selected TF genes, 25 candidates of each subset (CHS-group and FRK1-group) were subjected together with the respective marker genes to a GENEVESTIGATOR meta-analysis with the perturbation and biclustering tools [31]. All candidate TFs were analyzed additionally by an ATTED-II co-expression [32]. The log2-values from genes presented were compared to public transcriptomic data using the GENEVESTIGATOR platform and data sets AT-00107-4 (4 h), AT-00253-2 (3 h), and AT-00391-11 (2 h) were chosen for comparison with our flg22 single treatment. Though there were several other UV-B treatment experiments accessible, most dealt with plants grown on Murashige-Skoog (MS) medium supplemented with sucrose and might be therefore biased concerning FPG regulation. Thus, the comparison of our UV-B expression data with the public AT-00109-5 (UVAB treatment) and AT-00616-7 (low UV-B, but on MS) data sets is rather limited.

## 3. Results

### 3.1. RNAseq Data Reveals Transcriptomic Reprogramming in Plants by flg22 and UV-B as Well as Their Crosstalk/Interaction

To investigate the transcriptomic changes in response to flg22 treatment and UV-B irradiation-induced responses as well as their crosstalk/interaction, four treatments were conducted as described in M&M. In addition to a control (C), Arabidopsis seedlings were sprayed with 1 mM flg22 (F), exposed to UV-B (U), or co-treated with flg22 and UV-B (F/U). The pooled RNAs from each treatment were subjected to RNAseq to identify genes potentially involved in this crosstalk. A total of 24002 expressed genes were identified, however, 261 genes, which had an FPKM only in the control sample, were excluded from the analysis, leaving 23741 genes for DEG analysis (Appendix A). By comparative analysis, 10778, 13620, and 11294 DEGs were identified from flg22, UV-B, and F/U libraries, respectively (Figure 1A). There were more genes down-regulated than induced in all three treatments. In this study, we focused specifically on antagonistically regulated DEGs being co-regulated either with the *CHS* (CHS-group, being UV-B induced and suppressed by flg22) or reciprocally being co-regulated with the *FRK1* marker gene (FRK1-group, being flg22 induced and down-regulated by UV-B). From 7672 DEGs (Appendix A), 1915 and 1351 candidate DEGs were identified from CHS-group and FRK1-group, respectively (Figure 1B,C, Appendix A).

### 3.2. RNAseq Data Support Our Previously Postulated Working Model

Based on a KEGG enrichment analysis of DEGs from CHS-group and FRK1-group, there are several indications that the phenylpropanoid pathway (ath00940), including flavonoid biosynthesis (ath00941) and anthocyanin biosynthesis (ath00942), displays major changes in this crosstalk (Appendix A). Therefore, the hierarchical clustering of the relative gene expression (log2-fold change) of 45 DEGs encoding structural genes of the phenylpropanoid pathway was investigated in the context of this crosstalk. Generally, genes in Cluster I are suppressed by F or the co-treatment F/U and mostly up-regulated by UV-B (such as the marker gene *CHS*), while Cluster II genes are up-regulated by F or F/U, but negatively affected by UV-B treatment (such as the *FRK1* marker gene). The result is consistent with previous qPCR data from Arabidopsis cell cultures and Arabidopsis meta-data [5] (Figure 1D and Appendix A), demonstrating that flavonol biosynthesis pathway genes (FPGs) are generally suppressed by single F or F/U co-treatment in Arabidopsis plants. In addition, anthocyanidin biosynthetic-related genes *LDOX/ANS* and *DFR* showed a similar expression pattern as the FPGs. Here, the crosstalk marker gene *CHS* displayed the strongest induction by UV-B (log2 = 2.41), but three FPGs were found within Cluster II (*FFT*, *CHI,* and *UGT78D2*), while the *4CL3* was not detected in this data set. Instead, several lignin pathway genes (LPGs), such as *CAD2/3/6/7/9*, *LAC17,* and *F5H2* were assigned to Cluster I being suppressed by all three treatments. The only general phenylpropanoid pathway gene (GPP) found in Cluster I is the *PAL3* (phenylalanine ammonialyase), being indicative of metabolite channeling established during evolution by gene duplication [33]. The Arabidopsis single copy gene *C4H* (cinnamate-4-hydroxylase) and the remaining isoforms of PAL (PAL1/2/4) and 4CL (4CL1/2/5) [34] were much stronger induced by F and F/U than by UV-B, indicating distinct function from UV-protection and cluster together with the remaining genes involved in coumarin/scopoletin and/or lignin biosynthesis. *CCoAOMT6*, *CAD1*, *CAD8,* and *FRK1* are a few examples of genes that were strongly induced by both F and F/U but were slightly suppressed by UV-B radiation. In conclusion, RNAseq analysis revealed two distinct expression patterns, which separate FPGs and anthocyanin biosynthesis-related genes (APGs) that are involved e.g., in the production of UV-protective metabolites (Cluster I) from other branches of the phenylpropanoid pathway, which are important for pathogen defense such as lignin and phytoalexin biosynthesis (Cluster II).

### 3.3. New TFs Being Potentially Involved in This Crosstalk

In the next steps, we focused on our analysis of conversely regulated genes for TFs by the UV-B irradiation and by the flg22 treatment, respectively. In Appendix A, two major groups were defined from the DEG list based upon their co-regulation with the marker gene *CHS* (genes being UV-B up and down-regulated by both F and F/U) and with the PTI marker gene *FRK1* (genes being UV-B suppressed and up-regulated by F as well as F/U). In total, 112 and 107 TFs were identified for the CHS-group and FRK1-group, respectively (Table 1). From our previous study, several common *cis*-elements found in the promoters of the FPGs have been expected to be involved in transcriptional regulation of this crosstalk, comprising e.g., the MRE-box (ACCTACC bound by MYB TFs), ACE-box (CACGTg bound by bHLH or bZIP TFs), W-box (TTGACc bound by WRKY TFs) and the *FORC*^A^
*cis*-element (tTGGGC potentially targeted by NAC TFs). Therefore, the corresponding TFs were sought from the DEGs (Table 1 and Appendix A). 

As a result, 40 candidate TFs and WD40 proteins were found to display consistent regulatory patterns, 20 candidate TFs being co-regulated with the *CHS* (UV-B up-regulation and suppression by flg22) were presented in Table 2A, while the other 20 candidates are co-regulated with the *FRK1* (flg22 up-regulation and suppression by UV-B) were presented in Table 2B. These genes were compared to public transcriptomic data or biclustering using the GENEVESTIGATOR platform as shown in Appendix A, or the ATTED-II platform as shown in Appendix A (CHS-group) and Appendix A (FRK1-group). The potential links among the candidates are indicated in Table 2 (“linkage” column) based on the biclustering analysis (Appendix A) or putative regulatory hubs (Appendix A). The expression of these 40 candidates was further compared to experiments from public data sets accessed through GENEVESTIGATOR (Appendix A) showing in most cases similar tendencies.

### 3.4. Transcript Analysis of Selected Candidate Genes by RT-qPCR

To validate the RNAseq data, 6 random genes from Table 2A and Table 2B, respectively, together with the crosstalk marker genes *FRK1* and *CHS*, and several PTI-associated genes, such as structural genes for defense metabolite production (*F6H1*, *CCR2*, *HCT,* and *CCoAOMT6*), signaling components (BAK1, an integral part of flg22-sensing receptor complexes, MPK3 and MPK4), as well as PDF1.2, a plant defensin commonly used as a marker gene for jasmonate/ethylene mediated defense pathways were subjected to RT-qPCR analysis. Two known TFs being implicated in this crosstalk, MYB12, a positive regulator, and MYB4, a negative regulator, were also included in the analysis. Transcript analysis by RT-qPCR generally confirmed the RNAseq data. The single flg22 treatment as well as the co-treatment F/U led to a significant induction of FRK1-group genes and a suppression of CHS-group candidates (Figure 2). However, it was not possible to obtain statistically significant data demonstrating a negative effect of UV-B on the expression of flg22 up-regulated genes, most likely because these genes are expressed at a very low level and therefore a further reduction is difficult to observe or because PTI associated signaling components are post-translationally regulated [35], suggesting this crosstalk is predominantly uni-directional.

It is noted, that even though not detected in the RNAseq data, MYB4 (Figure 2L) and MYB12 (Figure 2D) showed a similar expression pattern as reported in our previous studies, implying a limitation of the RNAseq technology applied in this study.

## 4. Discussion

A transcriptomic approach was applied for identifying genes being involved in the crosstalk between flg22 and UV-B-induced signaling pathways. Both F and F/U treatments had similar strong effects on overall gene expression, while in the samples treated with moderate UV-B much fewer DEGs were detected (Figure 1A), reflecting that the plant response to biotic stress is much stronger and more complex than to abiotic UV-B stress. The expression patterns of several cell structural genes (Figure 1D and Appendix A) and TFs (Figure 2 and Appendix A), as well as the marker genes *CHS* and *FRK1* in our RNAseq data, are congruent to our previous study, which argues for the high quality of this RNAseq data set.

A subset of DEGs encoding TFs could be classified into two groups, being either up-regulated by UV-B and suppressed by single flg22 or the co-treatment flg22/UV-B (CHS-group) or down-regulated by UV-B and induced by single flg22 or the co-treatment flg22/UV-B (FRK1-group). Forty candidate TFs were chosen for further analysis (Table 2). Interestingly, most of these candidate genes from both groups displayed additional inverse regulation under several other stress treatments (Appendix A), as shown in a biclustering analysis using GENEVESTIGATOR (Appendix A). For instance, while CHS-group genes appear to be predominantly light-regulated (fitting to their strong induction by UV-B/VIS as applied in this study), their suppression was also observed under drought-inducing conditions (Appendix A). On the other hand, FRK1-group genes were generally up-regulated in response to biotic stress, but also in response to nitrogen depletion (Appendix A). 

In general, the genes being flg22-induced could be potential negative regulators of the FPGs, while flg22-suppressed genes might play a positive role in FPG regulation or negatively regulate pathogen defense genes. Thus, the selected candidates have the potential to contribute to this crosstalk. 

### 4.1. Possible Involvement of MBW Complexes

Both positive and negative regulating MYBs are implicated in the regulation of the “late genes” required for anthocyanin biosynthesis as part of MBW complexes, consisting of bHLH TFs and WD40 proteins [9,36]. The fact that several MYB, bHLH, and WD40 candidates were identified implies a possible role of MBW complexes in the FPG regulation as well. This is further supported by the identification of corresponding *cis*-elements in the promoters of FPGs, as well as promoter deletion studies [5,37]. Three positive regulators of FPGs MYB12, MYB75, and MYB111 are known co-regulators with the CHS (Figure 2A–D). Especially the anthocyanidin positive regulator, MYB75 [19], showed a strong down-regulation in the F and F/U treatments clustering with the CHS (Appendix A), suggesting the PTI signaling could negatively regulate these TFs to suppress FPG expression. 

The Arabidopsis genome contains 133 MYB TFs [38], 162 bHLH TFs [39], and 108 WD40 repeat containing proteins [40], indicating a great potential for functional redundancy in the composition of MBW complexes. To date, TTG1 is the only WD40 protein implicated in flavonoid biosynthesis [41]. The UV-B response depends on upstream components such as UVR8. UVR8 interaction with COP1 is negatively regulated by RUP1 and RUP2 and the latter three proteins all contain WD40-domains as well [42]. It is noted that COP1 was shown to mediate MYB75 degradation in the dark [43]. We speculate that some of the differentially regulated TF candidates might form previously undescribed MBW complexes and function either positively (consisting e.g., of MYB75-SPCH-AT5G53500) or negatively consisting, e.g., of MYB7/MYB32-bHLH78-AT1G64610. MYB4 and the phylogenetically related MYB7, are negative regulators of flavonoid production [44], while MYB32 is a negative regulator of lignin biosynthetic genes [45]. All were found to be co-regulated with the FRK1 (Figure 2I–L). However, other combinations could be deduced from the CHS-group, for example, an MBW complex containing negative regulators such as MYB29 and the bHLH TF UNE10, or one probably involved in trichome development consisting of MYB23, the bHLH TF MYC1, and the WD40 protein FAS2. Thus, it is reasonable to assume that the “early” FPGs are regulated by MBW complexes as suggested by Liu et al. [9] and indicated in Figure 3. Further experiments are needed, e.g., by mutant analysis, to ascertain the role of MBW complexes in crosstalk regulation.

### 4.2. Function of Genes Co-Regulated with the UV-B-Marker CHS

Two TF genes, *MYB23* and *ERF38*, were identified (Appendix A), and MYB23 is involved in trichome initiation which is positively regulated by UV-B. In addition to their repelling function towards insects, trichomes are also thought to protect the shaded tissue from UV-B radiation [46,47]. ERF38 is considered a candidate regulator of secondary wall metabolism in cell types that are not reinforced by the typical deposition of lignin but display suberization [48]. While ERF38 has not been linked to trichome development, several other TFs identified in Table 2A are involved in this process (GTL1, MYC1, and the WD40 protein FAS2). 

MYB111 was distantly linked to SPCH, MYC1, and HB31 (Appendix A). The positive regulator MYB111 has already been linked to this crosstalk as capable to activate *CHS* expression [6,19]. SPCH and MYC1 are both bHLH TFs, with SPCH being involved in stomatal development and indirectly stabilized by HY5-mediated expression of STOMAGEN [49,50]. Its suppression during PTI might be of advantage when reducing potential entry points of pathogens, but there is no link to UV-B-related responses so far. 

Appendix A links bZIP61, bZIP34, GTL1 and HAT1. Both bZIP61 and bZIP34 have been found to form heterodimers with bZIP18 and bZIP52 [51] and bind to the G-Box [52], but all these TFs have not been reported in the regulation of FPG expression before. GTL1, a trihelix TF, acts to interfere with trichome cell differentiation [53] and the trichome branch length [54]. HAT1 might function in a negative feedback mechanism, e.g., via interaction with MYB75, thereby interfering with MBW protein complex formation to stop the expression of the FPGs [55]. 

We found several interesting candidates, including UNE10, a bHLH TF, and the three Arabidopsis Whirly TFs, which cluster together with MYB29 and MYB75 (Appendix A). Both MYB29, as well as UNE10, are negative regulators and the latter can inhibit Phytochrome A-mediated far-red light responses [56]. UNE10 is one of three TFs that were stronger induced by UV-B, with an additional attenuating effect on flg22-mediated suppression observed in the co-treatment (Table 2A, together with MYB111 and SPCH). Co-regulation of CHS with the Whirly TFs constitutes another link to chloroplast and mitochondrial retrograde signaling. WHY1 is targeted to the chloroplast, while WHY2 is targeted to the mitochondria [57]. Interestingly, Lepage et al. [58] showed that anthocyanin levels are strongly suppressed in the *why1/why3/polIb* triple mutant. The suppression of WHY1, WHY2, and WHY3 during elicitor-induced PTI can be deduced (Appendix A). In the *why1* mutant, H_2_O_2_ and SA were induced earlier during senescence development [59,60], and immunity-related genes such as PR-1 were activated [61]. Thus, the suppression of Whirly TFs found in this study might be important for the integrity of flg22-induced plant immunity.

### 4.3. Role of Genes Co-Regulated with the PTI-Marker FRK1

The closely related TFs MYB7 and MYB32 are similar to MYB4 transcriptional repressors (Appendix A) [62], implying an extensive functional redundancy explaining why a single *myb4* knock-out mutant has no significant effect on this crosstalk. The anthocyanin repressor MYBL2 [63,64] was only weakly co-regulated with the FRK1. Therefore, it might be interesting to generate and analyze a triple *myb4/myb7/myb32* KO mutant to see if transcriptional suppression of FPGs is mediated by these MYBs together. 

The bHLH78 TF (AT5G48560) and WRKY9, as well as another bHLH TF (AT1G05710) with unknown functions, and WRKY39, were identified by ATTEDII analysis (Appendix A). WRKY9 and bHLH78 are linked via WRKY36, which is co-regulated with the FRK1 (Appendix A). WRKY36 usually acts to suppress HY5 transcription, but in response to UV-B, this TF interacts with UVR8 thereby releasing from the HY5 promoter [65]. WRKY9 is reported to regulate suberin production thereby forming another protective barrier and enhancing salt tolerance [66]. The biclustering analysis identified several negative and positive regulators belonging to the NAC and WRKY TF families (Appendix A). The negative regulators induced during PTI are potentially involved in the suppression of the FPGs, where they could bind to FPG promoters, or alternatively, regulate the TFs forming MBW complexes as indicated in the lower part of Figure 3. However, the negative regulators among them could also suppress the expression of TFs forming positive regulating MBW complexes in the upper part of Figure 3. NAC2 can be induced by salt, abscisic acid (ABA), the ethylene precursor ACC, and the synthetic auxin NAA [67], while NAC003 has been reported to be a negative regulator in vascular stem cell maintenance [68]. Interestingly, NAC053 (NTL4), a positive regulator, functions in regulating drought-induced leaf senescence [69] and programmed cell death under heat stress conditions [70]. NAC053 is furthermore transcriptionally regulated under mitochondrial stress conditions and probably binds to a mitochondrial dysfunction motif (CTTGNNNNNCA[AC]G), the motif TG[TG]CGTA or a Proteasome-Related *cis*-Element with the consensus core TGGGC [69,71,72], which was described as *FORC*^A^ Motif that responds to the crosstalk between defense and light signaling and found in all “early FPGs” [5,73]. WRKY6 has been shown to negatively regulate Phosphate1 (Pho1) expression in response to low phosphate (Pi) stress, resulting in increased anthocyanin production [74,75], but also low nitrogen availability can induce flavonoid production [76,77]. Moreover, WRKY75 RNAi plants were also susceptible to Pi stress as indicated by the higher accumulation of anthocyanin during Pi starvation, suggesting that this negative regulator might suppress FPG or APG expression [78].

## 5. Conclusions

Taken together, our transcriptomic analysis provided further evidence for the impaired production of UV-B protective flavonols and anthocyanins during activated PTI in favor of lignin/scopoletin formation. This report now constitutes a basis for further hypotheses-driven elucidation of the molecular mechanism(s) underlying this crosstalk. Some new hypotheses are summarized in Figure 3 and constitute a valuable dataset and framework for further research. For instance, yeast two-hybrid or split YFP experiments could be applied to verify the interaction between the TFs mentioned in the model. Understanding how plants integrate different environmental cues to adjust their metabolism is critical for improving crop varieties concerning nutritional quality and adaptation to increasing stress conditions caused e.g., by climate change. Whole transcriptome profiling is one possible strategy to identify differentially expressed candidate genes, which could be then manipulated in order to prove their involvement in the plants’ stress response.

## Figures and Tables

**Figure 1 genes-14-00240-f001:**
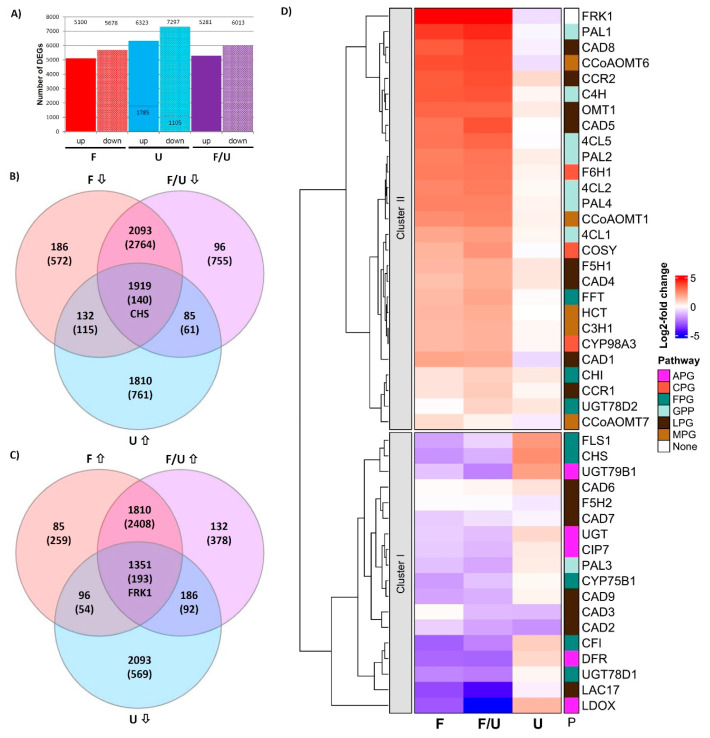
Impact of crosstalk on transcriptomic changes in Arabidopsis. (**A**) An overview of DEGs identified from RNAseq experiments using pooled RNA of three independent biological replicates. Treatments with flg22 (F), UV-B (U), or co-treatment with flg22 and UV-B (F/U) as compared to the control. DEGs were identified by applying a log2 threshold ≤ −1 or ≥ 1 for each treatment. Since we detected approximately five-fold fewer DEGs in the UV-B samples we lowered the log2 threshold to ±0.2 for this treatment in order to obtain a comparable number of DEGs as in the other treatments. (**B**) A Venn diagram displaying the number and distribution of DEGs that were down-regulated in response to single and co-treatment flg22, but up-regulated by UV-B (CHS-group). (**C**) A Venn diagram displaying the number and distribution of DEGs that were up-regulated in response to single and co-treatment with flg22, but down-regulated by UV-B (FRK1-group). The DEG numbers shown in (**B**,**C**) are based on a log2 threshold ≤−1 or ≥1 for flg22 or F/U co-treatment with the log2 threshold for the UV-B treatment lowered to ±0.2. (**D**) Cluster analysis of genes involved in phenylpropanoid metabolism shows the expected expression patterns with genes involved in the flavonoid pathway (FPG and anthocyanin biosynthesis-related genes) being largely up-regulated by UV-B and suppressed in response to flg22 and containing the *CHS* marker gene (Cluster I) as well as genes being up-regulated in response to flg22 (Cluster II), belonging to the general phenylpropanoid (GPP), monolignol (MPG), lignin (LPG) or coumarin (CPG) pathways clustering together with the *FRK1* marker gene for successful induction of pattern triggered immunity (PTI). Expression levels are calculated based on log2-fold changes relative to the untreated control. Gene name abbreviations and concrete log2 values underlying the calculation can be found in Appendix A.

**Figure 2 genes-14-00240-f002:**
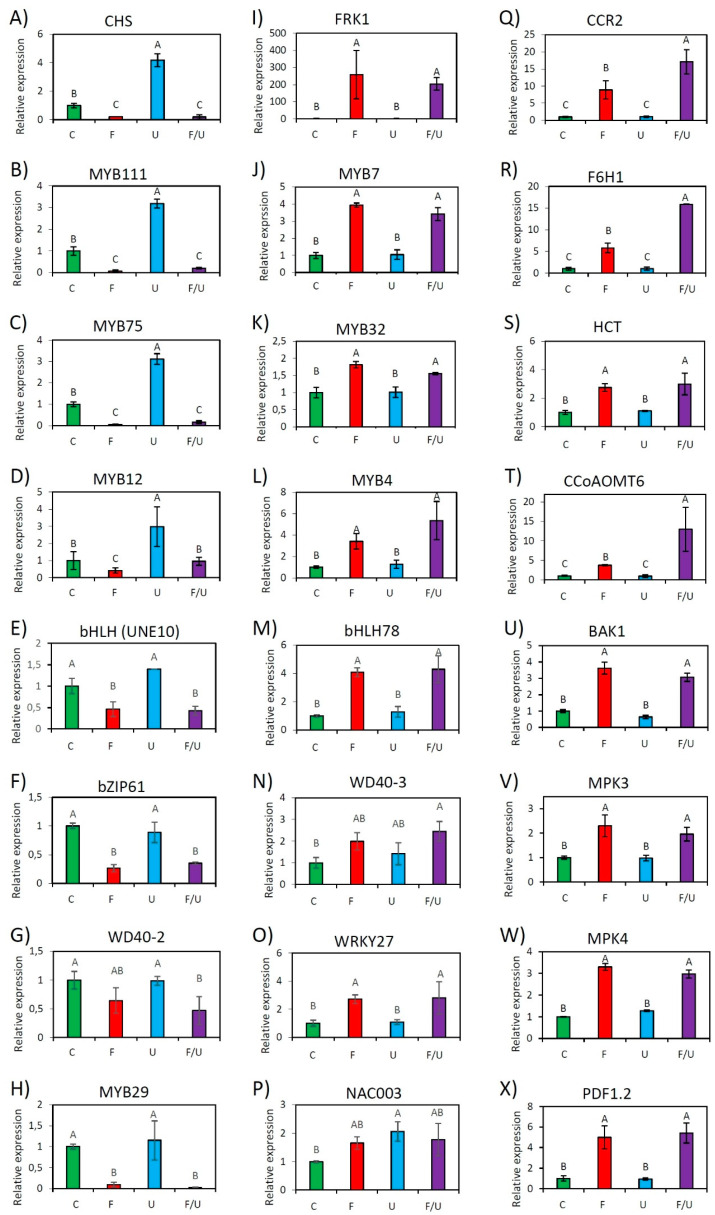
Relative expression levels of 24 selected genes from all identified clusters as analyzed by RT-qPCR. Shown is the response to different treatments with water as control (C), flg22 (F), UV-B (U), or co-treatment with flg22 and UV-B (F/U). The expression was measured after 4 h treatment of wild-type Arabidopsis Col-0 and is generally consistent with the Cluster analysis. Genes (**A**–**H**) display a reduction in response to flg22 single or co-treatment (Cluster I genes/CHS-group), while genes (**I**–**X**) are inducible by flg22 single or co-treatment (Cluster II genes/FRK1-group). Error bars represent the standard error of triplicate experiments, and statistical significance with *p* ≤ 0.05 was checked by a two-way ANOVA.

**Figure 3 genes-14-00240-f003:**
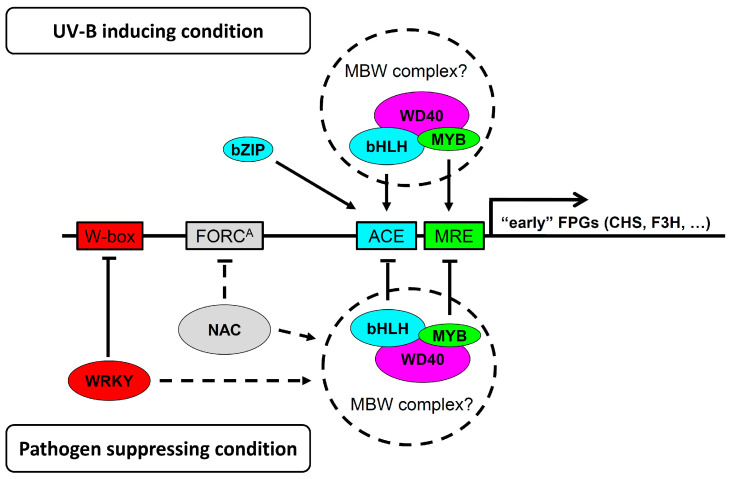
Graphical summary of potentially new processes involved in crosstalk between pathogen and UV-B induced signaling as discussed in the text. Identification of several conversely regulated MYB, bHLH, and WD40 TFs (Table 1) suggests that MBW complexes consisting of these TFs might also be regulating the “early” FPGs. Positive influence on FPG expression is deduced from genes that are co-regulated together with the *CHS* marker gene (being UV-B up- and flg22 down-regulated), while the negative impact is assumed from genes that are co-regulated with the *FRK1* marker gene (being UV-B down- and flg22 up-regulated).

**Table 1 genes-14-00240-t001:** Overview of some transcription factor families identified under the eight possible stress treatment outcomes.

Stress Outcome	DEGs	TFs	bHLH	MYB	NAC	WRKY	bZIP	Figure
U↑F↓F/U↓(CHS)	1919	112	11	9	2	1	3	Figure 1B
U↓F↑F/U↑(FRK1)	1351	107	5	14	10	17	7	Figure 1C
U↑F↓F/U↑	132	12	-	-	3	-	-	Appendix A
U↓F↑F/U↓	96	3	-	-	1	-	-	Appendix A
U↑F↑F/U↓	58	4	-	-	-	1	-	Appendix A
U↓F↓F/U↑	186	22	2	2	2	-	1	Appendix A
U↑F↑F/U↑	1810	171	14	18	13	19	5	Appendix A
U↓F↓F/U↓	2093	218	19	23	11	4	5	Appendix A
Total	7672	649	51	66	42	42	21	

**Table 2 genes-14-00240-t002:** Log2-values of top 20 candidate genes potentially involved in crosstalk between UV-B signaling and PTI.

**A: DEGs being co-regulated with the marker gene *CHS***						
**AGI Code**	**TF Type**	**Description**	**Linkage**	**flg22** **↓**	**UVB** **↑**	**F/U** **↓**
AT5G40330	MYB23	regulation of trichome initiation	Appendix A	−2.03	0.28	−0.90
AT5G07690	MYB29	Neg. regulator of mitochondrial stress responses	Appendix A	−3.39	0.66	−3.58
AT1G56650	MYB75	PAP1, regulation of anthocyanins / ROS scavenging	Appendix A	−4.11	0.94	−3.43
AT5G49330	MYB111	redundantly regulates flavonol biosynthesis	Appendix A	−7.26	1.60	−0.87
AT1G33240	MYB-like	GTL1, activator, trichome morphogenesis	Appendix A	−1.13	0.20	−1.08
AT4G00050	bHLH	UNE10 (PIF8), inhibits far-red light responses	Appendix A	−1.28	0.74	−0.33
AT5G53210	bHLH	SPCH, positive regulator, stomatal development	Appendix A	−1.46	0.71	−0.16
AT4G00480	bHLH12	MYC1, trichome patterning	Appendix A	−2.31	0.62	−3.21
AT3G58120	bZIP61	heterodimers with bZIP18/52, binds to G-boxes	Appendix A	−3.13	0.70	−3.89
AT2G42380	bZIP34	heterodimers with bZIP18/52, pos. regulator	Appendix A	−4.06	0.61	−3.64
AT5G64630	WD40-1	FAS2, trichome differentiation		−3.50	0.58	−2.07
AT5G53500	WD40-2	response to light stimulus, but cytoplasmatic protein	Appendix A	−2.11	0.59	−1.58
AT4G32980	ATH1	homeobox gene 1, involved in photomorphogenesis		−2.62	0.50	−2.10
AT1G14440	HB31	Homeobox protein, long-day photoperiodism	Appendix A	−2.51	0.23	−4.05
AT4G17460	HAT1	class II Homeodomain-ZIP protein, negative regulator	Appendix A	−1.59	0.35	−1.59
AT2G35700	ERF38	secondary cell wall biogenesis	Appendix A	−1.72	0.31	−1.99
AT1G04250	IAA17	AXR3, repressor of auxin-inducible gene expression	Appendix A	−2.15	1.11	−1.47
AT1G14410	WHY1	defense, retrograde signaling in chloroplasts	Appendix A	−1.32	0.32	−1.62
AT1G71260	WHY2	defense response, chloroplast, mitochondrion	Appendix A	−1.69	0.47	−1.99
AT2G02740	WHY3	defense response, chloroplast, mitochondrion	Appendix A	−1.01	0.37	−0.76
**B: DEGs being co-regulated with the marker gene *FRK1***						
**AGI Code**	**TF Type**	**Description**	**Linkage**	**flg22** **↓**	**UVB** **↓**	**F/U** **↓**
AT2G47190	MYB2	pos. regulator, ABA responsive, phosphate starvation	Appendix A	2.22	−0.32	2.42
AT2G16720	MYB7	repressor of flavonol biosynthesis, linked to MYB4	Appendix A	2.59	−0.35	3.32
AT4G34990	MYB32	Related to MYB3/4/7, response to light stimulus	Appendix A	1.78	−0.21	1.72
AT4G12350	MYB42	activates phenylalanine biosynthesis/lignin		3.30	−0.94	3.67
AT1G18570	MYB51	indole glucosinolate biosynthesis, callose, bacteria		2.36	−0.37	1.55
AT5G48560	bHLH78	response to blue light, CRY2-INTERACTING BHLH 2	Appendix A	1.40	−0.69	1.55
AT1G05710	bHLH	unknown function	Appendix A	1.54	−0.46	1.52
AT2G36270	bZIP	ABI5, positive regulator, response to ABA		1.98	−0.48	1.96
AT3G54620	bZIP25	pos. regulator		1.97	−0.37	1.87
AT2G46270	bZIP	GBF3, induced by ABA, cold, drought, and darkness		1.67	−0.53	1.55
AT1G64610	WD40-3	response to oxidative stress, defense to pathogen	Appendix A–E	1.82	−0.26	1.54
AT3G45620	WD40-4	defense response to bacterium, signal transduction		1.31	−0.45	1.31
AT1G62300	WRKY6	neg. regulator, response to phosphate starvation	Appendix A–F	4.05	−0.34	3.65
AT1G68150	WRKY9	chemical homeostasis, defense response	Appendix A	3.61	−2.21	2.97
AT5G52830	WRKY27	neg. regulator, defense response to bacterium		2.80	−1.17	2.46
AT3G04670	WRKY39	response to salicylic acid, developmental processes	Appendix A	1.40	−0.48	1.65
AT5G13080	WRKY75	neg. regulator, induced by Pi starvation	Appendix A–F	5.26	−0.24	5.43
AT5G39610	NAC2(6)	pos. regulator, ABA + ET responsive, involved in PCD	Appendix A	2.58	−0.73	2.73
AT1G02220	NAC003	neg. regulator, ABA-activated signaling pathway	Appendix A	3.69	−0.31	3.91
AT3G10500	NAC053	pos. regulator, ROS production	Appendix A	1.63	−0.44	1.63

## Data Availability

The data reported in this paper have been deposited in the National Center for Biotechnology Information Sequence Read Archive (NCBI SRA) Gene Expression Omnibus (GEO) database, https://www.ncbi.nlm.nih.gov/geo (accessible on 16 August 2021) (BioProject ID: PRJNA755194).

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
