# Peer review of "RNAseq-Based Working Model for Transcriptional Regulation of Crosstalk between Simultaneous Abiotic UV-B and Biotic Stresses in Plants"

_genes, 2023, doi:10.3390/genes14020240_

Round 1

Reviewer 1 Report

The Submitted manuscript entitled “RNAseq-based working model for transcriptional regulation of crosstalk between simultaneous abiotic UV-B and biotic stresses in plants” reports a crosstalk between signalling cascades triggered by two stressors simultaneously in plants. The study suggest that the interaction between UV-B and flg22 stress triggers extensive reprogramming of gene transcription involving a large number of different families of transcription factors. Moreover, the MBW complex may be part of the regulatory machinery and play a key role in the regulation of early FPG. The identification and characterization of the master-switch genes involved in this crosstalk is therefore of scientific and practical importance. 

Although the hypothetical model seems reasonable, it is necessary to correspondingly simplify and put this model into more relative terms because it is based only on RNAseq experiments. In addition, the strategy for validating this model should be explained in more detail.

Author Response

Thank you very much for your comments. According to your suggestion we have now simplified the model in Fig. 3 and explained a suitable strategy to validate the model in the future within the conclusion part (lines 461 - 462).

Reviewer 2 Report

The presented work is very interesting and it is very relevant in the present context of changing climate. It's a good attempt by the authors to understand the molecular mechanism regulating the crosstalk between abiotic and biotic stress in a model plant. Few suggestions are mentioned below that can be incorporated in the MS:

In the introduction section, few lines can be added on the negetive effects of UV-B on crops and how it is going to impact crop production in the face of changing climate. This will help the reader to understand why UV-B in particular has been used as the abiotic  stressor. Besides, it will strengthen the significance of the work presented.

Materials and method

Line 136-146: clarify whether stability assessment has been done for the housekeeping gene used in the study in the test condition in Arabidopsis? If so, provide relevant reference.  Also include the primer sequence of the housekeeping gene. 

Conclusion need to be more precise and with a simple take home message. 

Author Response

Thank you very much for your comments. According to your suggestion we  have added additional information about UVB effects in the introduction (lines 57 – 65). 

About the housekeeping gene:

Actin2 is a reliable and traditional reference gene in Arabidopsis (Fallath et al. 2017; Zhao et al., 2017; Stephan et al., 2019), which is used as an internal control to normalize gene expression in many studies. We regularly check its stability of expression in all 4 treatments by RT-PCR, which showed very similar expression pattern in different treatments, demonstrating it can used as a reliable housekeeping gene. Primer info including sequence and product length can be found in supplementary table S1.

Fallath, T., Kidd, B.N., Stiller, J., Davoine, C., Björklund, S., Manners, J.M., Kazan, K. and Schenk, P.M., 2017. MEDIATOR18 and MEDIATOR20 confer susceptibility to Fusarium oxysporum in Arabidopsis thaliana. PLoS One, 12(4), p.e0176022.Zhao, Y., Bi, K., Gao, Z., Chen, T., Liu, H., Xie, J., Cheng, J., Fu, Y. and Jiang, D., 2017. Transcriptome analysis of Arabidopsis thaliana in response to Plasmodiophora brassicae during early infection. Frontiers in microbiology, 8, p.673.Stephan, L., Tilmes, V. and Hülskamp, M., 2019. Selection and validation of reference genes for quantitative Real-Time PCR in Arabis alpina. PLoS One, 14(3), p.e0211172.

We have highlighted changes in the Conclusions section and added requested information (lines 461 – 467).